# Five-Membered Nitrogen Heterocycles Angiotensin-Converting Enzyme (ACE) Inhibitors Induced Angioedema: An Underdiagnosed Condition

**DOI:** 10.3390/ph17030360

**Published:** 2024-03-10

**Authors:** Niki Papapostolou, Stamatios Gregoriou, Alexander Katoulis, Michael Makris

**Affiliations:** 1Allergy Unit, 2nd Department of Dermatology and Venereology, National and Kapodistrian University of Athens, University General Hospital ‘Attikon’, 12462 Athens, Greece; alexanderkatoulis@yahoo.co.uk; 21st Department of Dermatology and Venereology, “Andreas Syggros” Hospital for Skin and Venereal Diseases, National and Kapodistrian University of Athens, 16121 Athens, Greece; stamgreg@yahoo.gr

**Keywords:** angiotensin-converting enzyme (ACE) inhibitors, angioedema, bradykinin, nitrogen-based heterocycles, pharmacology

## Abstract

Angiotensin-converting enzyme (ACE) inhibitors are used primarily in the treatment of hypertension, heart failure, and in the acute phase of myocardial infarction. Lisinopril [N^2^-[(1S)-1-car-boxy-3-phenylpropyl]-L-lysyl-L-proline], enalapril [(S)-1-[N-[1-(ethoxycarbonyl)-3-phenylpropyl]-L-alanyl]-L-proline] and ramipril [2-aza-bicyclo-[3.3.0]-octane-3-carboxylic acid] are all five-membered heterocycles and three of the most prevalent ACE inhibitors in clinical use worldwide. ACE inhibitor-induced angioedema (AE) is clinically characterized by self-limited edema of the dermis and subcutaneous lipid tissue, localized on face skin, oral mucosa and tongue in most cases. However, severe episodes of intestinal AE misdiagnosed as acute appendicitis and laryngeal AE requiring incubation have been reported. The pathophysiology of ACE inhibitor-induced angioedema is attributed to the accumulation of bradykinin, which is a potent vasodilator with proinflammatory activity that is normally degraded by angiotensin-converting enzyme (ACE) and aminopeptidase P; however, a small proportion of treated patients is affected. Given that patients do not respond to anti-H1 antihistamines and steroids, early clinical recognition and discontinuation of the ACE inhibitors are the treatments of choice for the long-term management of ACE inhibitor- induced angioedema. The search period of the present review was set up until November 2023, and its aim is to shed light on the broader context of ACE inhibitor-induced angioedema, exploring aspects such as clinical presentation, pathophysiology, and therapeutic considerations in this potentially life-threatening condition. The exploration of alternative drug options such as angiotensin II receptor blockers, the potential association of coadministration of DPP-4 inhibitors with ACE inhibitors, the presentation of angioedema and the significant clinical importance of this condition are also discussed. By focusing on the chemical structure of ACE inhibitors, specifically their nitrogen-based heterocycles—an attribute shared by over 880 drugs approved by the FDA within the pharmaceutical industry—this review emphasizes the pivotal role of nitrogen scaffolds in drug design and underscores their relevance in ACE inhibitor pharmacology.

## 1. Introduction

Nitrogen heterocycles play a pivotal role in the design and development of pharmaceutical agents, particularly in the field of angiotensin-converting enzyme (ACE) inhibitors [1]. ACE inhibitors are crucial components in the management of arterial hypertension and various cardiovascular conditions [2]. Among these inhibitors, a class of compounds with five-membered nitrogen heterocycles has garnered significant attention for their pharmacological properties, with enalapril [(S)-1-[N-[1-(ethoxycarbonyl)-3-phenylpropyl]-L-alanyl]-L-proline], ramipril [2-aza-bicyclo-[3.3.0]-octane-3-carboxylic acid] and lisinopril [N^2^-[(1S)-1-car-boxy-3-phenylpropyl]-L-lysyl-L-proline] representing the most prevalent ACE inhibitors in clinical use. These nitrogen-containing rings contribute to the structural diversity and bioactivity of ACE inhibitors, influencing their efficacy in modulating the renin-angiotensin-aldosterone system [3]. 

However, these medications, are not without their complexities, and a lesser-known but potentially serious side effect has been emerging—angioedema induced by five-membered nitrogen heterocycles ACE inhibitors [4]. It is worth noting that ACE inhibitor-induced AE is not restricted to the five-membered nitrogen heterocycle drugs of this category but is an adverse event in all ACE inhibitors. However, five-membered nitrogen heterocycles ACE inhibitors are the most widely used worldwide according to the literature, and it is proposed in the literature that they are more effective. In fact, in the USA, lisinopril is the most prescribed ACE inhibitor, while in Europe, there is a prevalence for ramipril, both being five-membered nitrogen heterocycles [5]. ACE inhibitors are the leading course of drug-induced angioedema in the USA [6]. ACE inhibitors induce angioedema in 0.1–0.7% of recipients, and although the rate is relatively low, the wide use of these drugs, with more than 40 million patients receiving them on a daily basis worldwide, makes them the most common cause of drug-induced angioedema [7,8,9,10]. The disease is not restricted to one ethnic group; however, ethnicity seems to play a significant role. People of African and Hispanic descent have an increased risk of ACE-induced angioedema [5,7]. Specifically, the incidence of ACE inhibitor-induced angioedema is up to five times greater in people of African descent [11]. However, a case–control study demonstrated that ACE inhibitors and black ethnicity do not seem to have a synergistic action (OR 1.10, 95% CI 0.80–1.51) [12]; thus, black ethnicity per se predisposes patients to experience angioedema attacks, regardless of the type of antihypertensive used [13]. 

Angioedema manifests as an asymmetrical, nonpitting swelling in subcutaneous or submucosal tissues, typically impacting areas not dependent on gravity. In cases of angioedema induced by ACE inhibitors, a distinctive feature is the absence of itching or urticaria; the presence of urticaria indicates different underlying causes. The most affected regions in ACE inhibitor-induced angioedema include the lips, tongue, face, and upper airway. While the involvement of the intestine can result in acute abdominal pain accompanied by diarrhea or other gastrointestinal symptoms, this manifestation may not be as readily recognized. The specific reason why these particular anatomical sites are more susceptible to involvement remains unknown [14]. The exact mechanisms behind ACE inhibitor-induced angioedema are complex and not fully understood. The pathophysiology of ACE-induced angioedema is attributed to the accumulation of bradykinin, which is a potent vasodilator with proinflammatory activity that is normally degraded by angiotensin-converting enzyme (ACE), aminopeptidase P (APP), neutral endopeptidase (NEP), dipeptidyl peptidase-4(DDP-4), and kininase I. The five-membered nitrogen heterocycles in certain ACE inhibitors may enhance bradykinin release, leading to increased vascular permeability and subsequent angioedema [15]. 

The search period of the present review was set up until November 2023. The objective of this review is to illuminate various facets of ACE inhibitor-induced angioedema within a broader context. It delves into the clinical manifestations, underlying pathophysiology, and treatment considerations of this potentially life-threatening condition. Additionally, alternative pharmacological options such as angiotensin II receptor blockers and the potential implications of coadministration of DPP-4 inhibitors with ACE inhibitors are explored. This review also discusses the presentation of angioedema and underscores its clinical significance. Through an examination of the chemical structure of ACE inhibitors, particularly their nitrogen-based heterocycles, which are a common feature among over 880 FDA-approved drugs, this review highlights the crucial role of nitrogen scaffolds in drug design and their significance in ACE inhibitor pharmacology.

## 2. Clinical Presentation

Angioedema is characterized by an asymmetrical, non-pitting swelling in subcutaneous or submucosal tissues, predominantly affecting areas not influenced by gravity [16]. When triggered by ACE inhibitors, a notable feature is the absence of itching or urticaria, with the presence of urticaria suggesting diverse underlying causes [17]. ACE inhibitor-induced angioedema typically impacts the lips, tongue, face, and upper airway [18]. Although the involvement of the intestine can lead to acute abdominal pain, diarrhea, or other gastrointestinal symptoms, this manifestation might not be immediately discernible [19]. Additionally, there have been rare cases of laryngeal angioedema requiring intubation [18]. There may be differences in the distribution of affected areas, with women more commonly experiencing facial angioedema, while men may have involvement of the tongue or throat [20]. 

Although episodes of AE occur spontaneously and episodically, the pattern of AE follows a relatively predictable course, with swelling of the affected areas occurring over minutes to hours and then resolving during the next 24–72 h, with a complete resolution usually during 5 days or more [6]. If the episode of AE is not recognized or not attributed to ACE inhibitors despite its remission, both the recurrence rate and the severity of the subsequent episodes are unpredictable [21]. 

## 3. Time of Onset of AE

Although angioedema induced by ACE inhibitors can arise at any stage during treatment, occurring at a rate of 1 in 1000 cases, more than half of instances typically present within the initial week [7,22]. According to a significant retrospective analysis, over two-thirds of angioedema episodes occur within the first trimester of commencing ACE inhibitor therapy [23]. Notably, case reports have documented occurrences of angioedema several years post-treatment initiation, particularly with enalapril, with incidents reported 9 and 23 years after initiating treatment, even in patients lacking known risk factors [24,25]. Similarly, there have been reports of lisinopril-induced angioedema occurring after 11 years of treatment [26]. While the onset time of angioedema after initiating ACE inhibitor therapy varies among individuals irrespective of gender, some research suggests that women may experience angioedema earlier after beginning treatment compared to men [27]. 

## 4. Suggested Mechanisms

### 4.1. The Role of ACE in Renin Angiotensin Aldosterone Pathway and Degradation of Bradykinin

ACE, also referred to as kininase II, typically functions by converting angiotensin I to angiotensin II in response to decreased blood pressure. Angiotensin II, in turn, affects various pathways, resulting in elevated blood pressure. ACE inhibitors mitigate blood pressure by intervening in multiple pathways.

Specifically, ACE inhibitors hinder the enzyme ACE, lowering blood pressure by impeding ACE function and reducing angiotensin II levels. They impact both the renin-angiotensin-aldosterone (RAA) pathway and the degradation of bradykinin. The RAA cascade manages renal blood flow and blood pressure, involving the conversion of angiotensinogen to angiotensin I by renin in the kidney. Subsequently, the enzyme ACE in the lungs metabolizes angiotensin I into angiotensin II, a vasoconstrictor that activates angiotensin I and II receptors [28]. 

Initially, inhibition of angiotensin II production leads to vasodilation. In addition, impaired metabolism of bradykinin, an inflammatory vasoactive peptide that induces vasodilation of blood vessels primarily by acting on bradykinin 2 receptors, leads to the elevation of its levels and its breakdown products named des-Arg-BK. This results in an increasing release of nitric oxide and prostaglandins, resulting in vasodilation and, thus, hypotension [29,30]. 

### 4.2. The Role of ACE inhibitors in AE

The clinical manifestations of ACE inhibitor-induced angioedema are associated with elevated levels of bradykinin. ACE is the primary peptidase involved in bradykinin degradation. Bradykinin, a nine-amino-acid peptide, enhances capillary permeability and acts as a potent vasodilator. Bradykinin production occurs following the cleavage of the high-molecular-weight kininogen (HMWK) by kallikrein, resulting in the active form of bradykinin [15,31] (Figure 1).

Bradykinin is primarily metabolized by ACE, neutral endopeptidase (NEP), and aminopeptidase P (APP), and secondarily by dipeptidyl peptidase-4 (DPP-4) and kininase I; thus, it has a brief half-life of approximately 17 s. Des-Arg9-BK, an active bradykinin metabolite, is primarily formed due to the kininase I enzyme. The pharmacological activities of des-Arg9-BK are short-lived due to breakdown by ACE and APP [14] (Figure 1).

ACE inhibitor-induced angioedema is believed to result from defective degradation of at least three vasoactive peptides: bradykinin, des-Arg9-BK (a bradykinin metabolite), and substance P [15]. Normally, bradykinin is inactivated by ACE, APP, DPP-4, and NEP, as mentioned earlier. The APP-inactivated bradykinin metabolite, des-Arg9-BK, is also degraded by DPP-4. Substance P is primarily inactivated by the enzyme DPP-4, with ACE and NEP playing secondary roles. A decreased activity of DPP-4 correlated with a prolonged half-life of substance P, but only in the presence of ACE inhibition, suggesting a requirement for multiple enzyme defects to inhibit degradation [32]. 

When drug therapy inhibits ACE, the secondary bradykinin metabolic enzymes (APP, kininase I, NEP, and DPP-4) assume a relatively larger role in degrading bradykinin, des-Arg9-BK, and substance P. Thus, defects or deficiencies of these enzymes theoretically predispose patients to developing angioedema when taking an ACE inhibitor. In line with these mechanisms are data from a study that demonstrates decreased APP activity in the sera of 39 patients with a history of ACE inhibitor-induced angioedema, compared with 39 ACE inhibitor-exposed controls [33]. Additionally, about half of patients experiencing ACE inhibitor-induced angioedema also have an enzyme defect involved in des-Arg9-BK metabolism, leading to its accumulation when ACE is inhibited [34]. In addition, during an episode of angioedema due to the use of the ACE inhibitor captopril, case reports showed a 10-fold increase in bradykinin levels, returning to normal levels during remission [35]. 

In essence, elevated bradykinin levels trigger vasodilation and heightened vascular permeability in the postcapillary venules, facilitating plasma extravasation into the submucosal tissue, ultimately resulting in angioedema.

While this mechanism is consistent across genders, variations in susceptibility to angioedema may exist due to genetic, hormonal, or other factors that could differ between men and women [36]. 

## 5. Risk Factors

The occurrence of angioedema in individuals initiating ACE inhibitor treatment varies, influenced by various risk factors, including both genetic and environmental exposures.

Notably, people of African descent explore an increased susceptibility to ACE inhibitor-induced angioedema, experiencing up to five times more frequent episodes [37,38]. Certain genetic variations, especially polymorphisms in genes encoding aminopeptidase P (APP) and neutral endopeptidase (NEP), are more common in this demographic, potentially leading to reduced enzyme levels [33]. Studies have identified polymorphisms in the X-Prolyl Aminopeptidase 2 (XPNPEP2) gene, responsible for encoding the APP enzyme, which holds a central role in bradykinin degradation. Such polymorphisms can reduce enzyme activity, increasing susceptibility to ACE inhibitor-induced angioedema [39]. Given that XPNPEP2 is located on the X-chromosome, this association is more prevalent in men than in women [40]. Additionally, polymorphisms in exons and regulatory regions may further contribute to reduced APP enzyme activity [41]. Beyond genetic factors, various environmental factors can also contribute to angioedema presentation [42]. The OCTAVE study proposed female sex, seasonal allergies, age above 65 years, a history of previous angioedema episodes, and NSAID use as potential risk factors for ACE inhibitor-induced angioedema [38]. Smoking has also been suggested as a risk factor [43]. 

Recently, Maroteau et al. utilized whole exposome sequencing for the first time and identified an association between factor V (F5) variants, involved in blood coagulation, and ACE inhibitor-induced angioedema. Similar mutations on Factor 5 (F5), akin to those associated with Hereditary Angioedema type III (HAE III), may increase prekallikrein enzyme activity, leading to elevated bradykinin production [44]. Additionally, a meta-analysis of genome-wide association studies assessing genetic risk factors for ACE inhibitor-induced angioedema proposed, for the first time, the involvement of fibrinolysis in the development of angioedema. Additionally, it provided further evidence for the involvement of the bradykinin and coagulation systems in angioedema development [45]. 

Furthermore, the use of mTOR inhibitors such as sirolimus and everolimus in patients receiving ACE inhibitors has been linked to an increased incidence of angioedema. This may be attributed to the interference of immunosuppressants with bradykinin metabolism by reducing dipeptidyl peptidase-4 (DDP4) enzyme activity [46]. Similarly, co-administration of DDP4 inhibitors (gliptins) with ACE inhibitors has been linked to a ninefold higher risk of angioedema [47]. 

Individuals with underlying angioedema attributed to C1 inhibitor deficiency or dysfunction, whether hereditary or acquired, can develop bradykinin-mediated angioedema independently of ACE inhibitors. However, these patients may remain asymptomatic until exposed to ACE inhibitors, necessitating caution and consideration when prescribing this drug category [48]. Additionally, ACE inhibitor-induced cough, a well-recognized side effect of ACE inhibitors, has been suggested as a risk factor for ACE inhibitor-induced angioedema [48]. 

It is worth noting that diabetes mellitus is associated with a reduced risk of ACE inhibitor-induced angioedema [49]. 

## 6. Diagnosis

The diagnosis of ACE inhibitor-induced angioedema relies primarily on clinical suspicion. The classical clinical presentation includes angioedema in characteristic anatomical sites, the absence of accompanying hives, and a lack of responsiveness to antihistamines or corticosteroids in patients undergoing ACE treatment. Although the diagnostic criteria remain the same regardless of gender, clinicians should be aware of potential gender-specific differences in presentation [27]. Early discontinuation of ACE inhibitors is crucial and the episodes are expected to fade. However, the effect of discontinuation may only become apparent after months, as recurrent episodes may persist for months after discontinuation.

Bradykinin-mediated angioedema is often over-diagnosed in patients treated with ACEIs. A study conducted by Douillard et al. revealed that 41% of patients initially suspected of suffering from ACE inhibitor-induced angioedema were eventually diagnosed with mast cell-mediated angioedema after discontinuing the ACE inhibitors for six months [50]. 

Regarding laboratory testing, there are no definitive tests for diagnosing ACE inhibitor-induced angioedema. However, if an alternative diagnosis is suspected, such as a family history of angioedema, a history of malignancy, or a lymphoproliferative disorder, assessing C4 levels is a reasonable initial approach. Low levels may suggest the need for further evaluation, including measuring C1 levels and function, as well as C1q [16] (Table 1).

Distinguishing ACE inhibitor-induced angioedema from other types of angioedema, particularly those mediated by bradykinin, requires careful consideration. Moreover, healthcare professionals should be able to distinguish between bradykinin-mediated AE and mast-cell-mediated AE, identifying their distinct subtypes that necessitate varied management approaches and pose different prognoses.

### Mast-Cell-Mediated Versus Bradykinin-Mediated AE

Differentiating between bradykinin-mediated and mast cell-mediated angioedema poses a challenge due to their similar clinical presentation. However, since the treatment and the clinical course of these types of angioedema differ significantly, it is crucial to make an accurate diagnosis [51]. Mast cell mediated angioedema can occur either by IgE-dependent (Type I hypersensitivity reaction based on Gell and Coombs classification) or non-IgE-dependent mechanisms (direct mast cell and basophil activation, disruption of arachidonic acid pathway). Chronic Urticaria, a common condition affecting 1–2% of the general population, presents with AE as sole manifestation in 10% of patients, while angioedema co-exists with wheals in approximately 50–60% [52,53,54]. AE is more common in Chronic Spontaneous Urticaria (CSU) compared to inducible Urticaria (CIndU) [55], and is associated with longer disease duration and higher disease burden [56]. 

Mast-cell-induced angioedema is rapid in onset and it has a duration of usually less than 48 h, accompanied by pruritus in most cases; the presence of wheals is quite common. Antihistamines, corticosteroids, and adrenaline—in the presence of anaphylaxis—are the treatment of choice in this type of AE [57]. In contrary, bradykinin-mediated AE has a slower onset, usually it develops gradually over hours, and lasts more than 48–72 h. Pruritus may be absent and pain or burning sensation, tickling, or pain are more common [58]. In addition, vomiting and diarrhea along with severe abdominal pain may be present in bradykinin-mediated angioedema. Antihistamines, corticosteroids, and adrenaline usually are ineffective in this type of angioedema [59] (Table 2). Novel assays are emerging to differentiate between histamine-mediated and bradykinin-mediated AE, with threshold-stimulated kallikrein activity being explored as a potential marker [60]. 

Additionally, Tie-2, FAP-α, tPA, sE-selectin, and Ang-2 have been identified as promising biomarkers for differentiating between hereditary angioedema, ACE inhibitor-induced angioedema, and angioedema associated with chronic spontaneous urticaria [61]. 

## 7. Management

The initial approach to managing ACE inhibitor AE involves discontinuing the drug and, in cases of laryngeal AE, providing acute airway management. Although it remains unclear, gender-related factors could influence airway management and may need more aggressive interventions. As mentioned above, AE typically resolves within 24–72 h; however, if ACE inhibitors are continued, attacks may escalate in severity, posing potential dangers and life-threatening risks. Therefore, despite the lack of proven efficacy, exploring potential treatments is valuable in the management of ACE inhibitor angioedema.

▯Fresh frozen plasma (FFP)

Administration of FFP is thought to decrease bradykinin levels, as it contains ACE and other enzymes contributing to the degradation of bradykinin [62]. It is suggested that two units of plasma are adequate to resolve AE within 2 to 4 h [63]. However, it is important to note that besides ACE, FFP also contains HMWK and kallikrein, which might lead to bradykinin formation. This could potentially explain the rare instances of AE worsening following FFP administration [64]. 

▯Icatibant

Icatibant, an antagonist of bradykinin B2 receptors approved for treating HAE, has shown promise in ACE inhibitor-induced AE based on case reports indicating significant symptom resolution upon its administration in ACE inhibitor-induced AE cases [65,66]. A randomized study by Bas et al. further supported the use of icatibant over antihistamines or corticosteroids in ACEI-induced AE, revealing a median time to resolution of 8 h compared to 27.1 h, respectively [67]. However, two other randomized studies failed to replicate these results, thus leading to a debate on the efficacy of icatibant in ACE inhibitor-induced AE [68,69]. 

▯Ecallantide

Ecallantide, acting as a plasma kallikrein inhibitor, reduces the production of bradykinin. In a triple-blind, placebo-controlled trial comparing ecallantide to conventional therapy with a placebo, the primary endpoint focused on eligibility for discharge within 4 h of treatment. Ecallantide and the placebo both met the primary endpoint in 31% and 21% of patients, respectively, suggesting some treatment benefit with ecallantide [70]. However, in another randomized, double-blind, controlled trial involving the administration of a single subcutaneous dose of ecallantide, there was no significant reduction in time to discharge compared to the placebo [71]. 

▯C1 inhibitor concentrate

The fundamental abnormality in hereditary angioedema (HAE) types I and II lies in the deficiency of functional C1 inhibitor, a regulator of multiple proteases in the complement, contact system, coagulation, and fibrinolytic pathways. HAE can be effectively treated with administration of C1 esterase inhibitor [14]. It is assumed that, through its inhibitory action on contact system and kallikrein-mediated bradykinin generation, C1INH would downregulate kinin generation and facilitate overall bradykinin removal [72]. Positive results from some case reports support the use of C1 esterase inhibitor in ACE inhibitor-induced AE, although its application remains controversial [73,74]. In a proof-of-concept case series by Greve et al., C1 inhibitor reduced the time to complete symptom resolution to 10.1 h, compared to the standard of care with antihistamines and corticosteroids, which took 33.1 h. None of the patients treated with C1 inhibitor required intubation, while in the control group, three needed tracheotomy, and two required intubation [75]. However, in a double-blind, parallel group, multicenter randomized placebo-controlled trial of adult patients with ACE inhibitor-induced angioedema with airway obstruction, patients were randomized 1:1 for single doses of either C1INH or placebo (0.9% NaCl) intravenously, in addition to standard care. Surprisingly, the C1INH treatment was found to be inferior in the treatment of ACE inhibitor-induced angioedema compared to the placebo, specifically concerning the time to complete the resolution of symptoms (29.63 h ± 15.56 h in the C1INH arm and 17.29 h ± 10.40 h in the placebo arm) [76]. 

Further studies will be needed to demonstrate whether C1 inhibitor can be used effectively in ACE inhibitor-induced angioedema.

## 8. Prognosis

The main strategy in managing ACE inhibitor-induced angioedema is the early discontinuation of the drug, requiring early recognition and a high level of suspicion. In its typical clinical manifestation, angioedema resolves within 24–72 h. Continuing the drug poses an increased risk of life-threatening attacks. Importantly, angioedema may reappear, even after cessation of ACE inhibitors, and persist for months.

In a long-term follow-up involving 111 patients with ACE-related angioedema, 46% experienced recurrent episodes after discontinuation of ACE inhibitors. Notably, 88% reported a recurrence of angioedema within the initial month after treatment cessation, with only 2% encountering an episode after three months of discontinuation. This highlights the importance of vigilance and continuous monitoring even after ACE inhibitor discontinuation [77]. While the overall prognosis may be similar across genders, individual outcomes may vary based on factors such as comorbidities and response to treatment [7,78]. 

## 9. Future Perspectives: Related Drugs and AE

Exploring alternative drug options for patients experiencing ACE inhibitor-induced angioedema presents a fascinating path for future research. Angiotensin II receptor blockers (ARBs), known for their cardioprotective effects like ACE inhibitors, have drawn attention in this context. Earlier systematic reviews indicated an increased incidence of angioedema in patients transitioning from ACE inhibitors to ARBs [79,80]. However, recent studies have shown a lower frequency of angioedema in those specifically treated with ARBs [81]. It is important to consider that patients with a history of ACE inhibitor-induced angioedema may experience recurrent episodes during the period following ACE inhibitor discontinuation, which could potentially affect the observed outcomes when initiating ARBs. However, large-scale studies consistently show a decreased incidence of angioedema in patients receiving ARBs compared to other antihypertensive medications [82]. Another intriguing aspect is the potential association between angioedema and dipeptidyl peptidase-4 (DPP-4) inhibitors, commonly known as gliptins, which are commonly prescribed in the management of type 2 diabetes. While the prevalence of DPP-4 inhibitor-induced angioedema is not fully understood, these drugs are frequently prescribed alongside ACE inhibitors. This raises the possibility that angioedema could be attributed to the use of gliptins alone or in combination with ACE inhibitors. The underlying mechanism involves DPP-4 being one of the enzymes responsible for bradykinin degradation, and inhibiting these enzymes in susceptible individuals may lead to the development of angioedema. Further exploration of these scenarios is necessary to enhance our comprehension of potential alternatives in the management of ACE inhibitor angioedema [83]. 

## 10. Addressing ACE Inhibitor-Induced Angioedema through Tailored Interventions

Tailored interventions and personalized medicine hold immense promise in hypertension management, particularly in accurately predicting ACE inhibitor-induced angioedema. Developing a clinical tool to assess the risk of ACE inhibitor-induced angioedema could revolutionize clinical decision making, potentially saving lives by guiding clinicians to select alternative antihypertensive drugs that pose lower risks for susceptible patients. Incorporating genetic information into such assessments could be pivotal, highlighting the importance of further advancements in pharmacogenomics [36]. 

## 11. Cutting-Edge Horizons

Marching into the forefront of scientific exploration, recent advancements have unveiled unexplored pathways in ACE inhibition, surpassing traditional remedies and extending into interactions within cancer therapies.

Natural compounds, particularly metabolites from Traditional Chinese Medicine, have demonstrated the ability to inhibit angiotensin-converting enzymes, thereby playing a role in regulating blood pressure. As interest in herbal remedies for hypertension grows due to their perceived lower incidence of adverse effects, there is a rising trend among individuals to explore these traditional treatments. A recent study by Wu et al. pioneers the investigation into the interaction between ACEI-type herbal metabolites and ACE1 proteins, offering insights into the pharmacological mechanisms of action of each compound. Moreover, it lays the groundwork for refining herbal substances, a crucial step towards the development of reliable and effective ACE inhibitors in future research endeavors [84]. 

In addition, there is a growing interest in dietary compounds that could potentially act as effective ACE inhibitors. Among these compounds, peptides known for their antihypertensive properties have shown significant blood pressure-lowering effects with minimal toxicity. Considering this context, Yue et al. chose to study the dipeptide CW (with an IC50 of 0.16 μM), which has been identified for its potent ACE inhibition activity, and five distinct Metal-Organic Frameworks (MOFs) with varying pore structures, metal ions, and organic ligands as adsorbents to assess their capability to absorb the dipeptide CW. This study marks the first systematic exploration of the adsorption interactions between MOFs and ACE inhibition peptides, shedding light on their potential applications [85]. 

Recently, enzyme inhibition properties have been proposed from transition metal complexes derived from Schiff bases. Synthesized compounds with their observed enzyme inhibition properties, particularly the Ni(II)/Zn(II) complex, could potentially exhibit ACE inhibitory effects, making them relevant candidates for further exploration as cardiovascular drugs, including ACE inhibitors [86]. 

Regarding malignancies management, ACE inhibition has been noted to have implications in cancer treatment, particularly in the context of radiotherapy. By inhibiting ACE, the levels of reactive oxygen species (ROS) are lowered, consequently diminishing the sensitivity to radiotherapy in cases of nasopharyngeal carcinoma. These findings highlight that ACE inhibitors, such as enalaprilat, can confer resistance to ionizing radiation (IR) by reducing ROS levels within nasopharyngeal carcinoma cells [87]. 

## 12. Conclusions

In summary, this review underscores the critical role of nitrogen heterocycles in pharmaceutical design, particularly in the development of angiotensin-converting enzyme inhibitors crucial for managing hypertension and cardiovascular conditions. However, it highlights the lesser-known yet serious risk of angioedema induced by five-membered nitrogen heterocycles present in these medications.

Understanding the clinical presentation, pathophysiology, and treatment options for ACE inhibitor-induced angioedema is paramount due to its potential underdiagnosis and life-threatening nature. Recognition of the risk factors, including genetic considerations, is vital, and prompt discontinuation of ACE inhibitors are the primary interventions, despite diagnostic challenges. This review explores management strategies such as fresh frozen plasma, icatibant, ecallantide, and C1 inhibitor concentrate, and considers alternative drug options like angiotensin II receptor blockers and potential associations of DPP-4 inhibitors with ACE inhibitor-induced angioedema. It stresses the urgent need for clinicians to differentiate this condition from other forms of angioedema, particularly mast cell-mediated types, as early recognition and long-term monitoring are crucial given the possibility of recurrence even after discontinuation of ACE inhibitors.

While acknowledging limitations such as the lack of systematic methodology and the restricted search period, this review remains a valuable addition to the literature on ACE inhibitor-induced angioedema. By addressing unresolved questions and emphasizing existing knowledge, it sheds new light on drug design intricacies and pharmacological considerations while increasing clinical awareness of this life-threatening condition.

## Figures and Tables

**Figure 1 pharmaceuticals-17-00360-f001:**
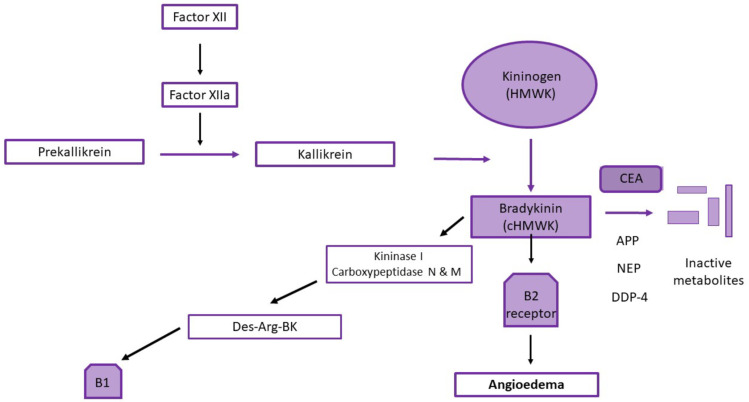
Pathways involved in bradykinin mediated angioedema.

**Table 1 pharmaceuticals-17-00360-t001:** Comparison of laboratory features of bradykinin mediated angioedema and mast cell mediated angioedema.

Angioedema Disorder	C4	C1 INH Levels	C1 Inhibitor Function	C1q	Other
HAE with C1inhibitor deficiency Type I	Low	Low	Low (usually < 50%)	Normal	
HAE with C1 inhibitor deficiency type II	Low	Normal or elevated	Low (usually < 50%)	Normal	
HAE with normal C1 inhibitor	Normal	Normal	Normal	Normal	
Acquired AE with C1 inhibitor Deficiency	Low	Normal or Low	Low (usually < 50%)	Normal or low	Anti C1 Inhibitor antibodies
ACEI- AE	Normal	Normal	Normal	Normal	
Mast cell mediated AE	Normal	Normal	Normal	Normal	

**Table 2 pharmaceuticals-17-00360-t002:** Comparison of clinical characteristics of mast cell mediated angioedema and bradykinin mediated angioedema. + : present. - : not commonly present and ++ : more commonly present.

	Mast Cell Mediated AE	Bradykinin Mediated AE (ACEIs-Induced AE)
Wheals/history of recurrent wheals	+	-
Recurrent Laryngeal AE	-	+
Tongue AE	+	++
Angioedema triggered by NSAIDs	+	-
Rate of onset	minutes	hours
Duration	<48 h	>48 h
Response to antihistamines/ corticosteroids/ adrenaline	+	-

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
