# Peer review of "Five-Membered Nitrogen Heterocycles Angiotensin-Converting Enzyme (ACE) Inhibitors Induced Angioedema: An Underdiagnosed Condition"

_pharmaceuticals, 2024, doi:10.3390/ph17030360_

Round 1

Reviewer 1 Report

Comments and Suggestions for Authors

The following are missing or not appropriate to the current review manuscript.

1. The Figures and tables are of low quality, and not much literature is cited under each section of chosen specific topic.

2. There is no mention of the period under which they have made the review. 

3. The abstract is unclear with respect to the objective of the review

4.  affiliation and key words are not proper

Due to the above mentioned reason, I must reject this manuscript to be published in this journal.  

Comments on the Quality of English Language

Extensive english edits required

Author Response

Response to Reviewer’s 1 comments

  • Response: Thank you for reviewing our manuscript. Even though you rejected the manuscript to be published in the journal, your comments are of great value for the improvement of our manuscript, and we have tried in this revised version to address them all. All changes are incorporated into the manuscript. Besides, English style and typos throughout the text were carefully checked as well as grammatical and spelling mistakes. However, if you still insist on extensive edits, we are more than willing to have the manuscript copy-edited professionally according to your suggestions. For your convenience, we uploaded a marked and unmarked form of the revised form.

Below we provide a point-by-point reply to your comments.

  1. The Figures and tables are of low quality, and not much literature is cited under each section of chosen specific topic.

  • Response: We have uploaded editable versions of tables and figures, and we are more than willing to modify them according to your suggestions. Also, more references were added in the text both in the introduction section and the rest of the subsections. Please find them below:

Reichman, M. E., Wernecke, M., Graham, D. J., Liao, J., Yap, J., Chillarige, Y., Southworth, M. R., Keeton, S., Goulding, M. R., Mott, K., & Kelman, J. A. (2017). Antihypertensive drug associated angioedema: effect modification by race/ethnicity. Pharmacoepidemiol Drug Saf, 26(10), 1190-1196. https://doi.org/10.1002/pds.4260

Mahmoudpour, S. H., Baranova, E. V., Souverein, P. C., Asselbergs, F. W., de Boer, A., & Maitland-van der Zee, A. H. (2016). Determinants of angiotensin-converting enzyme inhibitor (ACEI) intolerance and angioedema in the UK Clinical Practice Research Datalink. Br J Clin Pharmacol, 82(6), 1647-1659. https://doi.org/10.1111/bcp.13090

Kamil, R. J., Jerschow, E., Loftus, P. A., Tan, M., Fried, M. P., Smith, R. V., Foster, D., & Ow, T. J. (2016). Case-control study evaluating competing risk factors for angioedema in a high-risk population. Laryngoscope, 126(8), 1823-1830. https://doi.org/10.1002/lary.25821

Kostis, W. J., Shetty, M., Chowdhury, Y. S., & Kostis, J. B. (2018). ACE Inhibitor-Induced Angioedema: a Review. Current Hypertension Reports, 20(7), 55. https://doi.org/10.1007/s11906-018-0859-x

Bernstein, J. A., Cremonesi, P., Hoffmann, T. K., & Hollingsworth, J. (2017). Angioedema in the emergency department: a practical guide to differential diagnosis and management. International Journal of Emergency Medicine, 10(1), 15. https://doi.org/10.1186/s12245-017-0141-z

 Mathey, C. M., Maj, C., Eriksson, N., Krebs, K., Westmeier, J., David, F. S., Koromina, M., Scheer, A. B., Szabo, N., Wedi, B., Wieczorek, D., Amann, P. M., Löffler, H., Koch, L., Schöffl, C., Dickel, H., Ganjuur, N., Hornung, T., Buhl, T., Greve, J., Wurpts, G., Aygören-Pürsün, E., Steffens, M., Herms, S., Heilmann-Heimbach, S., Hoffmann, P., Schmidt, B., Mavarani, L., Andresen, T., Sørensen, S. B., Andersen, V., Vogel, U., Landén, M., Bulik, C. M., Bygum, A., Magnusson, P. K. E., von Buchwald, C., Hallberg, P., Rye Ostrowski, S., Sørensen, E., Pedersen, O. B., Ullum, H., Erikstrup, C., Bundgaard, H., Milani, L., Rasmussen, E. R., Wadelius, M., Ghouse, J., Sachs, B., Nöthen, M. M., & Forstner, A. J. (2024). Meta-analysis of ACE inhibitor-induced angioedema identifies novel risk locus. J Allergy Clin Immunol. https://doi.org/10.1016/j.jaci.2023.11.921

Rubin, S., & Tomaszewski, M. (2024). Prediction and prevention of ACE-inhibitor-induced angioedema—an unmet clinical need in management of hypertension. Hypertension Research, 47(2), 257-260. https://doi.org/10.1038/s41440-023-01491-9

Long, B. J., Koyfman, A., & Gottlieb, M. (2019). Evaluation and Management of Angioedema in the Emergency Department. West J Emerg Med, 20(4), 587-600. https://doi.org/10.5811/westjem.2019.5.42650

 Nia, A. M., & Er, F. (2013). Angioedema associated with the use of angiotensin-converting enzyme inhibitor. Cmaj, 185(1), E80. https://doi.org/10.1503/cmaj.111387

  1. There is no mention of the period under which they have made the review. 
  • Response: Thank you again for your comment. This review is not a systematic one. The search period was set up until November 2023. Pubmed, Google Scholar and Cochrane Database were searched to identify the relevant papers. To maximize the search coverage, a combination of medical subject heading (MeSH) terms for “angiotensin converting enzyme inhibitors”, “angioedema”, and “nitrogen-based heterocycles” were included. We added a comment regarding the period under which the review was made in line 73. We also rewrote the aim of the review in the introduction section in lines 80-88.

“The search period of the present review was set up until November 2023.The objective of this review is to illuminate various facets of ACE inhibitor-induced angioedema within a broader context. It delves into the clinical manifestations, underlying pathophysiology, and treatment considerations of this potentially life-threatening condition. Additionally, alternative pharmacological options such as angiotensin II receptor blockers and the potential implications of coadministration of DPP-4 inhibitors with ACE inhibitors are explored. The review also discusses the presentation of angioedema and underscores its clinical significance. Through an examination of the chemical structure of ACE inhibitors, particularly their nitrogen-based heterocycles, which are a common feature among over 880 FDA-approved drugs, the review highlights the crucial role of nitrogen scaffolds in drug design and their significance in ACE inhibitor pharmacology.”

  1. The abstract is unclear with respect to the objective of the review.

  • Response: We rewrote the abstract and added some more information regarding the aim of the review in lines 24-31.

“The present review aims to sheds light on the broader context of ACE inhibitor-induced angioedema, exploring aspects such as clinical presentation, pathophysiology, and therapeutic considerations in this potentially life-threatening condition. The exploration of alternative drug options such as angiotensin II receptor blockers, the potential association of coadministration of DPP-4 inhibitors with ACE inhibitors, the presentation of angioedema and the significant clinical importance of this condition are also discussed. By focusing on the chemical structure of ACE inhibitors, specifically their nitrogen-based heterocycles—an attribute shared by over 880 drugs approved by the FDA within the pharmaceutical industry—the review emphasizes the pivotal role of nitrogen scaffolds in drug design and underscores their relevance in ACE inhibitor pharmacology.”

  1. affiliation and key words are not proper
  • Response: We apologize for not including key words in this version of the manuscript. We have now added the following key words:

-Angiotensin- converting enzyme (ACE) inhibitors

-Angioedema

-bradykinin

-nitrogen-based heterocycles

-pharmacology

Regarding affiliations, they were automatically generated by the journal system and will be rectified accordingly.

Due to the above-mentioned reason, I must reject this manuscript to be published in this journal.  

Comments on the Quality of English Language

Extensive english edits required

  • Response: English style and typos throughout the text were carefully checked as well as grammatical and spelling mistakes. In case this is not adequate, we are more than willing to have the manuscript professionally copy-edited.

Reviewer 2 Report

Comments and Suggestions for Authors

It is well prepared but some suggestions were raised for improving the contents of paper.

1.       To date, there have been many papers of review on ACE inhibitors and angioedema. There have been case reports to alarm the underdiagnoses.  In this situation, what is the novelty of this review? Could you more stress and write focusing on the novelty as a review?

2.       So sorry; the conclusive epilogue may be the textbook sentence although this was related to the above situation. Could you write focusing more on the novelty?

3.       It is hard to read and follow the paper in some parts; for instance, the abbreviated word ‘ACE, ACEi and ACE-AE’ and fill-spelled words after abbreviating them were mixed. Also, ACEIs in row 32 could be already abbreviated ACE inhibitors in row 31 as ACEIs. In row 43, the word equivalent to ACEi-AE did not appear to be described. In row 62, ACE was abbreviated, but ACE was already abbreviated in row 31….

Comments on the Quality of English Language

Minor editing of English language required.

Author Response

Response to Reviewer’s 2 comments

It is well prepared but some suggestions were raised for improving the contents of paper.

  • Response: Thank you for your favorable consideration of our manuscript. All your comments and suggestions have been addressed and were of great value regarding the revision of our manuscript. Changes are incorporated into the manuscript. We uploaded a marked and unmarked form of our manuscript for your convenience. Below we provide point-by-point responses to each of your comments.

  1. To date, there have been many papers of review on ACE inhibitors and angioedema. There have been case reports to alarm the underdiagnoses.  In this situation, what is the novelty of this review? Could you more stress and write focusing on the novelty as a review?

  • Response: The present review sheds light on the broader context of ACE inhibitor-induced angioedema, exploring aspects such as clinical presentation, pathophysiology, and therapeutic considerations in this potentially life-threatening condition. While existing reviews on ACE inhibitor-induced angioedema exist, this review adds value to the literature by providing a comprehensive examination of various aspects and by enhancing clinical awareness while anticipating future research directions. The exploration of alternative drug options such as angiotensin II receptor blockers and the potential association of coadministration of DPP-4 inhibitors with ACE inhibitors and the presentation of angioedema are also discussed.

       Furthermore, the review underscores the significant clinical importance of this condition and clearly distinguishes mast cell-mediated angioedema from bradykinin-mediated angioedema induced by ACE inhibitors. The discussion of unresolved questions and the emphasis on existing knowledge, coupled with the analysis of current practices, could establish the text as a valuable contribution to the literature on ACE-induced adverse effects.

        The importance lies in the increased need for constant awareness of this condition, given that new genetic studies aiming to identify novel genetic factors are continually emerging, highlighting the significance of this entity. Additionally, by focusing on the chemical structure of ACE inhibitors, specifically their nitrogen-based heterocycles—an attribute shared by over 880 drugs approved by the FDA within the pharmaceutical industry—the text emphasizes the pivotal role of nitrogen scaffolds in drug design and underscores their relevance in ACE inhibitor pharmacology.

We tried to incorporate some of the above text in the abstract as well as in the introduction section in lines 24-31 and 55-62 respectively.

  1. So sorry; the conclusive epilogue may be the textbook sentence although this was related to the above situation. Could you write focusing more on the novelty?

  • Response: Thank you again for the focused observation. We rewrote the last section of the conclusion trying to focus on the novelty of our review and highlight its strengths in lines 403-407.

      “By addressing unresolved questions and emphasizing existing knowledge, this review emerges as a valuable addition to the literature on ACE inhibitor-induced angioedema. It sheds new light on the intricacies of drug design and pharmacological considerations, while also clearly differentiating between mast cell-mediated angioedema and bradykinin-mediated angioedema. Furthermore, it significantly increases clinical awareness of this life-threatening entity.”

  1. It is hard to read and follow the paper in some parts; for instance, the abbreviated word ‘ACE, ACEi and ACE-AE’ and fill-spelled words after abbreviating them were mixed. Also, ACEIs in row 32 could be already abbreviated ACE inhibitors in row 31 as ACEIs. In row 43, the word equivalent to ACEi-AE did not appear to be described. In row 62, ACE was abbreviated, but ACE was already abbreviated in row 31….

  • Response: We have now used “ACE” (Angiotensin Converting Enzyme) as the only abbreviation in the text to enhance readability.

Minor editing of English language required.

  • Response: English style and typos throughout the text were carefully checked as well as grammatical and spelling mistakes. We are more than willing to have the manuscript professionally copy-edited according to your suggestions.

Reviewer 3 Report

Comments and Suggestions for Authors

Major Comments:

The authors need to address the following queries:

1.   Please briefly explain the importance of the study.

2.   Please discuss about the prevalence of ACEi-AE globally. Is the disease restricted to specific ethnic group or race?

3.    If ACEi other than five-membered nitrogen heterocycles are effective enough, then the simple way out could be the avoidance of using five-membered nitrogen heterocycles containing ACEi. Please discuss.

4.   The authors may discuss about the gender specific variation on subsections 2 to 9 of the review.

Minor Comments:

1.     The authors should thoroughly check the manuscript for the grammatical errors.

Comments on the Quality of English Language

The authors should thoroughly check the manuscript for the grammatical errors.

Author Response

Response to Reviewer’s 3 comments

The authors need to address the following queries:

 Response: Thank you for your consideration of our manuscript. All your comments and suggestions have been addressed and were of great value regarding the revision of our manuscript. Changes are incorporated into the manuscript. We uploaded a marked and unmarked form of our manuscript for your convenience. Below we provide point-by-point responses to each of your comments.

  1. Please briefly explain the importance of the study.
  • Response: The present review sheds light on the broader context of ACE inhibitor-induced angioedema, exploring aspects such as clinical presentation, pathophysiology, and therapeutic considerations in this potentially life-threatening condition. While existing reviews on ACE inhibitor-induced angioedema exist, this review adds value to the literature by providing a comprehensive examination of various aspects and by enhancing clinical awareness while anticipating future research directions. The exploration of alternative drug options such as angiotensin II receptor blockers and the potential association of coadministration of DPP-4 inhibitors with ACE inhibitors and the presentation of angioedema are also discussed. Furthermore, the review underscores the significant clinical importance of this condition and clearly distinguishes mast cell-mediated angioedema from bradykinin-mediated angioedema induced by ACE inhibitors. The discussion of unresolved questions and the emphasis on existing knowledge, coupled with the analysis of current practices, could establish the text as a valuable contribution to the literature on ACE-induced adverse effects.

The importance lies in the increased need for constant awareness of this condition given that new genetic studies aiming to identify novel genetic factors are continually emerging, highlighting the significance of this entity. Additionally, by focusing on the chemical structure of ACE inhibitors, specifically their nitrogen-based heterocycles—an attribute shared by over 880 drugs approved by the FDA within the pharmaceutical industry—the text emphasizes the pivotal role of nitrogen scaffolds in drug design and underscores their relevance in ACE inhibitor pharmacology.

  1. Please discuss about the prevalence of ACEi-AE globally. Is the disease restricted to specific ethnic group or race?

  • Response: Thank you for your comment. We have expanded the brief mention of ACE inhibitor-induced angioedema (AE) in the introduction section and delved further into the global incidence and the influence of ethnic groups. The following paragraph now encompasses these points, and we have included three additional references (Lines 55-62)

“ACE inhibitors induce angioedema in 0.1-0.7% of the recipients and although the rate is relatively low, the wide use of these drugs with more than 40 million patients receiving them in daily basis worldwide, makes them the most common cause of drug-induced angioedema. The disease is not restricted to one ethnic group; however, ethnicity seems to play a significant role. People of African and Hispanic descent have an increased risk of ACE-induced angioedema. Specifically, the incidence of ACE inhibitor-induced angioedema is up to five times greater in people of African descent. However, a case-control study demonstrated that ACE inhibitors and black ethnicity do not seem to have a synergistic action (OR 1.10, 95% CI 0.80–1.51); thus, black ethnicity per se predisposes patients to experience angioedema attacks, regardless of the type of antihypertensive used.”

  1. If ACEi other than five-membered nitrogen heterocycles are effective enough, then the simple way out could be the avoidance of using five-membered nitrogen heterocycles containing ACEi. Please discuss.

  • Response: ACE inhibitor-induced AE is not restricted to the five-membered nitrogen heterocycles drugs of this category but is an adverse event in all ACE inhibitors. However, five-membered nitrogen heterocycles ACE inhibitors are the most widely used worldwide according to literature, and it is proposed in the literature to be more effective. In fact, in the USA, lisinopril is the most prescribed ACE inhibitor, while in Europe there is a prevalence for ramipril, both being five-membered nitrogen heterocycles.".(1) We have added the above paragraph in the text in lines 49-54.

  1. The authors may discuss about the gender specific variation on subsections 2 to 9 of the review.

  • Response: Thank you for your comment. Female gender is associated with an increased risk of ACE inhibitors induced angioedema as stated in lines 206. There are gender-specific variations in ACE inhibitor-induced angioedema across various aspects including clinical presentation, time of onset, pathomechanisms, diagnosis, management, and prognosis. Please find below the comments on subsections 2-8.

Subsection 2 lines 96-98: “There may be differences in the distribution of affected areas, with women more commonly experiencing facial angioedema while men may have involvement of the tongue or throat.”

Subsection 3 lines 115-117: “While the time of onset of angioedema after initiating ACE inhibitor therapy can vary among individuals regardless of gender, some studies indicate that women may develop angioedema earlier after starting treatment compared to men.”

Subsection 4 lines:176-177“While this mechanism is consistent across genders, variations in susceptibility to angioedema may exist due to genetic, hormonal, or other factors that could differ between men and women”.

Subsection 6 lines 228-230: “While the diagnostic criteria remain the same regardless of gender, clinicians should be aware of potential gender-specific differences in presentation.”

Subsection 7 lines 286-288: “Although it remains unclear, gender-related factors could influence airway management and may need more aggressive interventions.”

Subsection 8 lines 350-352: “While overall prognosis may be similar between genders, individual outcomes may vary based on factors such as comorbidities and response to treatment.”

Minor Comments:

  1. The authors should thoroughly check the manuscript for the grammatical errors.

  • Response: English style and typos throughout the text were carefully checked as well as grammatical and spelling mistakes. In case this is not adequate, we are more than willing to have the manuscript professionally copy-edited.

  1. Mahmoudpour SH, Baranova EV, Souverein PC, Asselbergs FW, de Boer A, Maitland-van der Zee AH. Determinants of angiotensin-converting enzyme inhibitor (ACEI) intolerance and angioedema in the UK Clinical Practice Research Datalink. Br J Clin Pharmacol. 2016;82(6):1647-59.

Round 2

Reviewer 1 Report

Comments and Suggestions for Authors

The authors have made edits to the manuscript, which is better than the earlier version. However, the following needs to be addressed

1.      The authors need to mention the coverage period in the abstract.

2.      The following references are worth mentioning in connection with the review: a) https://doi.org/10.3390/molecules28207131  b) https://doi.org/10.1016/j.seppur.2023.124620 c) https://doi.org/10.1016/j.molstruc.2022.134384  d) https://doi.org/10.3390/biomedicines11061581

3.      The limitations and the author's perspective should be mentioned at the end.

4.      The similarity report % match is too high (26%). Preferably, it should be less than 10%

Comments on the Quality of English Language

The authors need to revise the manuscript to reduce the matching percentage of the similarity report of iThenticate.

Author Response

Response to Reviewer’s 1 comments

The authors have made edits to the manuscript, which is better than the earlier version. However, the following needs to be addressed.

  • Response: Thank you again for reviewing our manuscript, even though you initially rejected it. Your comments are of great value for the constant improvement of our manuscript, and we have tried once again to address them all.

Below we provide a point-by-point reply to your comments.

  1. The authors need to mention the coverage period in the abstract.
  • Response: The coverage period is now mentioned in the abstract in lines 24-26:

“The search period of the present review was set up until November 2023, and its aim is to shed light on the broader context of ACE inhibitor-induced angioedema, exploring aspects such as clinical presentation, pathophysiology, and therapeutic considerations in this potentially life-threatening condition.”

  1. The following references are worth mentioning in connection with the review: a) https://doi.org/10.3390/molecules28207131  b) https://doi.org/10.1016/j.seppur.2023.124620c) https://doi.org/10.1016/j.molstruc.2022.134384  d) https://doi.org/10.3390/biomedicines11061581
  • Response: the above mentioned references have now been incorporated into the manuscript in a new subsection 11 in lines 378-401.

“11. Cutting-Edge Horizons

   Marching into the forefront of scientific exploration, recent advancements have unveiled unexplored pathways in ACE inhibition, surpassing traditional remedies and extending into interactions within cancer therapies.

   Natural compounds, particularly metabolites from Traditional Chinese Medicine, have demonstrated the ability to inhibit angiotensin-converting enzymes, thereby playing a role in regulating blood pressure. As interest in herbal remedies for hypertension grows due to their perceived lower incidence of adverse effects, there is a rising trend among individuals to explore these traditional treatments. A recent study by Wu et al. pioneers the investigation into the interaction between ACEI-type herbal metabolites and ACE1 proteins, offering insights into the pharmacological mechanisms of action of each compound. Moreover, it lays the groundwork for refining herbal substances, a crucial step towards the development of reliable and effective ACE inhibitors in future research endeavors. (Wu Q, Jiao Y, Luo M, Wang J, Li J, Ma Y, et al. Detection of Various Traditional Chinese Medicinal Metabolites as Angiotensin-Converting Enzyme Inhibitors: Molecular Docking, Activity Testing, and Surface Plasmon Resonance Approaches. Molecules. 2023;28(20):7131.)

    In addition, there is a growing interest in dietary compounds that could potentially act as effective ACE inhibitors. Among these compounds, peptides known for their antihypertensive properties have shown significant blood pressure-lowering effects with minimal toxicity. Considering this context, Yue et al. chose to study the dipeptide CW (with an IC50 of 0.16 μM), which has been identified for its potent ACE inhibition activity, and five distinct Metal-Organic Frameworks (MOFs) with varying pore structures, metal ions, and organic ligands as adsorbents to assess their capability to absorb the dipeptide CW. This study marks the first systematic exploration of the adsorption interactions between MOFs and ACE inhibition peptides, shedding light on their potential applications. (Yue S, Shao S, Mu G, Jalil Shah S, Yu X, Sun W, et al. Highly selective and pH responsive adsorption of ZIF-8 for angiotensin-converting enzyme (ACE) inhibitory active peptides and its mechanism. Separation and Purification Technology. 2023;324:124620.)

   Recently enzyme inhibition properties have been proposed from transition metal complexes derived from Schiffs bases. Synthesized compounds with their observed enzyme inhibition properties, particularly the Ni(II)/Zn(II) complex, could potentially exhibit ACE inhibitory effects, making them relevant candidates for further exploration as cardiovascular drugs, including ACE inhibitors. (Krishna GA, Dhanya TM, Shanty AA, Raghu KG, Mohanan PV. Transition metal complexes of imidazole derived Schiff bases: Antioxidant/anti-inflammatory/antimicrobial/enzyme inhibition and cytotoxicity properties. Journal of Molecular Structure. 2023;1274:134384)

   Regarding malignancies management, ACE inhibition has been noted to have implications in cancer treatment, particularly in the context of radiotherapy. By inhibiting ACE, the levels of reactive oxygen species (ROS) are lowered, consequently diminishing the sensitivity to radiotherapy in cases of nasopharyngeal carcinoma. These findings highlight that ACE inhibitors, such as enalaprilat, can confer resistance to ionizing radiation (IR) by reducing ROS levels within nasopharyngeal carcinoma cells. (Ding Y, Xiu H, Zhang Y, Ke M, Lin L, Yan H, et al. Learning and Investigation of the Role of Angiotensin-Converting Enzyme in Radiotherapy for Nasopharyngeal Carcinoma. Biomedicines. 2023;11(6):1581.)”

  1. The limitations and the author's perspective should be mentioned at the end.
  • Response: We rewrote the conclusion section in order to incorporate the limitations of the study and our perspectives. Please find the new conclusion in lines 407-423.

“In summary, this review underscores the critical role of nitrogen heterocycles in pharmaceutical design, particularly in the development of angiotensin-converting enzyme inhibitors crucial for managing hypertension and cardiovascular conditions. However, it highlights the lesser-known yet serious risk of angioedema induced by five-membered nitrogen heterocycles present in these medications.

   Understanding the clinical presentation, pathophysiology, and treatment options for ACE inhibitor-induced angioedema is paramount due to its potential underdiagnosis and life-threatening nature. Recognition of risk factors, including genetic considerations, is vital, and prompt discontinuation of ACE inhibitors is the primary intervention despite diagnostic challenges. The review explores management strategies such as fresh frozen plasma, icatibant, ecallantide, and C1 inhibitor concentrate, and considers alternative drug options like angiotensin II receptor blockers and potential associations of DPP-4 inhibitors with ACE inhibitor-induced angioedema. It stresses the urgent need for clinicians to differentiate this condition from other forms of angioedema, particularly mast cell-mediated types, as early recognition and long-term monitoring are crucial given the possibility of recurrence even after discontinuation of ACE inhibitors.

   While acknowledging limitations such as the lack of systematic methodology and the restricted search period, this review remains a valuable addition to the literature on ACE inhibitor-induced angioedema. By addressing unresolved questions and emphasizing existing knowledge, it sheds new light on drug design intricacies and pharmacological considerations while increasing clinical awareness of this life-threatening condition.”

 “The present review emphasizes the clinical importance of ACE induced angioedema entity, and focuses on therapeutic considerations, highlighting the need for early recognition of the entity as discontinuation of the ACE inhibitor is the treatment of choice for long-term management of ACE inhibitor induced angioedema.By focusing on the chemical structure of ACE inhibitors, particularly their nitrogen-based heterocycles, the review also highlights the role of nitrogen scaffolds in drug design and their relevance in ACE-inhibitor pharmacology.
   The primary limitations of this review are, firstly, its lack of systematic nature, and secondly, the search period. Research beyond November 2023 might not have been included, potentially causing it to miss recent advancements in understanding or managing ACE inhibitor-induced angioedema. Additionally, the review may not cover all possible aspects of ACE inhibitor-induced angioedema due to its specific focus on clinical presentation, pathophysiology, and therapeutic consideration
s.”

  1. The similarity report % match is too high (26%). Preferably, it should be less than 10%
  • Response: Thank you for addressing the issue with the similarity index. We have made efforts to reduce it, and we think that it is now close to 10%. Unfortunately, we do not have access to iThenticate to calculate the exact matching percentage. We are more than happy to make further changes according to your suggestions.

Reviewer 3 Report

Comments and Suggestions for Authors

The manuscript is now improved sufficiently.

Author Response

Thank you for favorably considering our manuscript. Your comments were of great value in revising it.